# jVMC: Versatile and performant variational Monte Carlo leveraging automated differentiation and GPU acceleration

Markus Schmitt[1][*] and Moritz Reh[2]

**1** Institut für Theoretische Physik, Universität zu Köln, Köln, Germany
**2** Kirchhoff-Institut für Physik, Universität Heidelberg, Heidelberg, Germany
* markus.schmitt@uni-koeln.de

December 15, 2021

## Abstract

The introduction of Neural Quantum States (NQS) has recently given a new twist to variational Monte Carlo (VMC). The ability to systematically reduce the bias of the wave function ansatz renders the approach widely applicable. However, performant implementations are crucial to reach the numerical state of the art. Here, we present a Python codebase that supports arbitrary NQS architectures and model Hamiltonians. Additionally leveraging automatic differentiation, just-in-time compilation to accelerators, and distributed computing, it is designed to facilitate the composition of efficient NQS algorithms.

## 1 Introduction

The numerical simulation of strongly correlated quantum many-body systems constitutes a major challenge in computational physics. Even when fully exploiting modern supercomputers, direct solutions that involve the complete state vector are restricted to a few dozen degrees of freedom, because the dimension of the underlying Hilbert space grows

exponentially with system size [1]. Therefore, suited approximate methods are required in order to address large or infinite system sizes, which are typically of interest when studying universal behavior. Particularly desirable are controlled methods, which can become numerically exact by tuning a control parameter.

During the past decades the development of a number of powerful numerical methods enabled substantial advances of our understanding of quantum many-body systems in and out of equilibrium. Prime examples are tensor network algorithms for low-dimensional systems [2–4], dynamical mean field theory in high dimensions [5–8], and quantum Monte Carlo [9] for equilibrium properties in the absence of a sign problem. However, despite the broad applicability of these methods, various physical problems remain elusive for state of the art numerical approaches, among which, e.g., systems in two or three spatial dimensions, long time dynamics, and frustrated systems. Since the limitations of the established methods in these regards are well understood, novel paths need to be explored to address the existing challenges.

The recent proposal of neural quantum states (NQS) as a new ansatz class for variational quantum Monte Carlo opened new perspectives and holds the potential to overcome existing limitations [10]. With their expressive power and their applicability independent of specific spatial structures NQS are a suited basis to devise a new family of broadly applicable algorithms. Their utility has been outlined in a number of fundamental works. It has been shown that NQS can be used to describe ground states of critical systems in two dimensions [11] or states with (chiral) topological order [12–14] and their suitability to study frustrated magnets is under investigation [15–19]. Moreover, various approaches have been developed for the simulation of non-equilibrium dynamics in closed [10, 20–26] and open quantum systems [27–29]. Other directions include the incorporation of NQS in quantum chemical simulations [30–32], quantum state tomography [33, 34], and the simulation of quantum computation [35].

A key practical aspect that NQS algorithms have in common with many other deep learning applications is the possibility to harness cutting edge supercomputing resources at large scales; in fact, exploiting the intrinsic parallelism is crucial to achieve state-of-the-art results. In this paper we introduce the jVMC Python codebase that provides efficient implementations of the typical tasks that will be at the core of any algorithm involving NQS, namely composing and evaluating arbitrary network architectures, sampling, and evaluating quantum operators. The implementations and the interfaces are designed to enable the straightforward composition of custom algorithms and the utilization of distributed compute clusters, ideally with GPU or TPU accelerators. For this purpose the code is based on the JAX library [36], that provides functionality for automatic differentiation and optimized just-in-time compilation targeted at the available compute resources. Distributed computing across multiple nodes is enabled by incorporating the Message Passing Interface (MPI) through the `mpi4py` library [37, 38].

The repository containing the jVMC source code and a detailed documentation are available online [39, 40]. The codebase and all dependencies can be installed as a Python package via

```
>>> pip install jVMC
```

## 2 Variational Monte Carlo algorithms

In this section we outline common Variational Monte Carlo (VMC) algorithms in order to highlight what are the core building blocks for which the jVMC codebase provides

solutions. For this purpose we discuss how to find ground states and low-lying excited states and how to simulate real time dynamics of isolated or open systems using a time-dependent variational principle (TDVP).

For simplicity we will consider spin-1/2 degrees of freedom and corresponding computational basis configurations $\mathbf{s} = (s_1, \ldots, s_N)$ with $s_i = \uparrow, \downarrow$. The generalization to higher-dimensional local Hilbert spaces is straightforward.

## 2.1 Quantum expectation values

The starting point of variational Monte Carlo is a variational ansatz for the coefficients $\psi_\theta(\mathbf{s})$ with variational parameters $\theta = (\theta_1, \ldots, \theta_K)$, such that without further assumptions the generally unnormalized wave function reads

$$|\psi(\theta)\rangle = \sum_{\mathbf{s}} \psi_\theta(\mathbf{s}) |\mathbf{s}\rangle \ . \tag{1}$$

In the following we assume that $\theta_k \in \mathbb{R}$. Given a wave function in this form the expectation value of any quantum operator $\hat{O}$ can be rewritten as

$$\frac{\langle \psi(\theta)|\hat{O}|\psi(\theta)\rangle}{\langle \psi(\theta)|\psi(\theta)\rangle} = \sum_{\mathbf{s},\mathbf{s}'} \frac{\psi_\theta(\mathbf{s})^* \psi_\theta(\mathbf{s}')}{\langle \psi(\theta)|\psi(\theta)\rangle} \langle \mathbf{s}|\hat{O}|\mathbf{s}'\rangle = \sum_{\mathbf{s}} \frac{|\psi_\theta(s)|^2}{\langle \psi(\theta)|\psi(\theta)\rangle} \sum_{\mathbf{s}'} \langle \mathbf{s}|\hat{O}|\mathbf{s}'\rangle \frac{\psi_\theta(\mathbf{s}')}{\psi_\theta(\mathbf{s})} \ . \tag{2}$$

Since $p_\theta(\mathbf{s}) \equiv |\psi_\theta(\mathbf{s})|^2 / \langle \psi(\theta)|\psi(\theta)\rangle$ constitutes a probability distribution on the computational basis configurations, we can write

$$\frac{\langle \psi(\theta)|\hat{O}|\psi(\theta)\rangle}{\langle \psi(\theta)|\psi(\theta)\rangle} = \sum_{\mathbf{s}} p_\theta(\mathbf{s}) O_{\text{loc}}^\theta(\mathbf{s}) \equiv \langle\!\langle O_{\text{loc}}^\theta \rangle\!\rangle_\theta \tag{3}$$

where we introduced the local estimator

$$O_{\text{loc}}^\theta(\mathbf{s}) = \sum_{\mathbf{s}'} \langle \mathbf{s}|\hat{O}|\mathbf{s}'\rangle \frac{\psi_\theta(\mathbf{s}')}{\psi_\theta(\mathbf{s})} \tag{4}$$

and the notation $\langle\!\langle \cdot \rangle\!\rangle_\theta$ for an expectation value with respect to $p_\theta(\mathbf{s})$. Since typical quantum operators are sparse in the computational basis, the sum $\sum_{\mathbf{s}'}$ in Eq. (4) can be evaluated efficiently.

By contrast, the sum over all basis states, $\sum_{\mathbf{s}}$, in Eq. (3) becomes unfeasible for large system sizes $N$, because the Hilbert space dimension grows exponentially with $N$. Therefore, one has to resort to Monte Carlo (MC) sampling of $p_\theta(\mathbf{s})$ to efficiently estimate these expectation values; for this purpose, only the functional form of $\psi_\theta(\mathbf{s})$ must allow for efficient evaluation [41]. Then samples $\{\mathbf{s}^{(1)}, \ldots, \mathbf{s}^{(N_{\text{MC}})}\}_{\mathbf{s} \sim p_\theta(\mathbf{s})}$ can be obtained using, e.g., the Metropolis-Hastings algorithm, and the expectation value can be approximated by the empirical mean

$$\langle\!\langle O_{\text{loc}}^\theta \rangle\!\rangle_\theta \approx \frac{1}{N_{\text{MC}}} \sum_{n=1}^{N_{\text{MC}}} O_{\text{loc}}^\theta(\mathbf{s}^{(n)}) \ . \tag{5}$$

## 2.2 Finding low-energy states

Given a variational ansatz $\psi_\theta(\mathbf{s})$ the ground state of a many-body Hamiltonian $\hat{H}$ can be approximated variationally using gradient-based optimization. The goal is to find the minimal energy expectation value

$$E(\theta) = \frac{\langle \psi(\theta)|\hat{H}|\psi(\theta)\rangle}{\langle \psi(\theta)|\psi(\theta)\rangle} \tag{6}$$

that is permitted by the chosen ansatz. The gradient with respect to the variational parameters $\theta_k$ takes the form

$$\partial_{\theta_k} E(\theta) = 2\mathrm{Re}\left( \sum_{\mathbf{s}} p_\theta(s)\left[\mathcal{O}_k^\theta(\mathbf{s})\right]^* E_{\mathrm{loc}}^\theta(\mathbf{s}) - \sum_{\mathbf{s}} p_\theta(\mathbf{s})\left[\mathcal{O}_k^\theta(\mathbf{s})\right]^* \sum_{\mathbf{s}} p_\theta(\mathbf{s}) E_{\mathrm{loc}}^\theta(\mathbf{s}) \right)$$

$$= 2\mathrm{Re}\left( \left\langle\!\!\left\langle \left(\mathcal{O}_k^\theta\right)^* E_{\mathrm{loc}}^\theta \right\rangle\!\!\right\rangle_\theta - \left\langle\!\!\left\langle \left(\mathcal{O}_k^\theta\right)^* \right\rangle\!\!\right\rangle_\theta \left\langle\!\!\left\langle E_{\mathrm{loc}}^\theta \right\rangle\!\!\right\rangle_\theta \right) \equiv 2\mathrm{Re}\left( \left\langle\!\!\left\langle \left(\mathcal{O}_k^\theta\right)^* E_{\mathrm{loc}}^\theta \right\rangle\!\!\right\rangle_\theta^c \right) \tag{7}$$

Here, $E_{\mathrm{loc}}^\theta(\mathbf{s}) = \sum_{\mathbf{s}'} \langle \mathbf{s}|\hat{H}|\mathbf{s}'\rangle \frac{\psi_\theta(\mathbf{s}')}{\psi_\theta(\mathbf{s})}$ as in Eq. (4) is the local energy. Moreover, we introduced the logarithmic derivatives

$$\mathcal{O}_k^\theta(\mathbf{s}) = \partial_{\theta_k} \log \psi_\theta(\mathbf{s}) \ , \tag{8}$$

which appear due to a multiplication by a unity $\psi_\theta(\mathbf{s})/\psi_\theta(\mathbf{s})$ in order to obtain factors of $|\psi_\theta(\mathbf{s})|^2$ analogous to Eq. (4). Finally, we use the notation $\langle\!\langle AB \rangle\!\rangle^c \equiv \langle\!\langle AB \rangle\!\rangle - \langle\!\langle A \rangle\!\rangle \langle\!\langle B \rangle\!\rangle$ for connected correlation functions.

The energy gradient (7) is again an expectation value with respect to $p_\theta(\mathbf{s})$, which we can estimate by MC sampling. Therefore, we can set up an iterative procedure to minimize the energy by updating the variational parameters according to the gradient descent prescription

$$\theta_k^{(n+1)} = \theta_k^{(n)} - \tau \partial_{\theta_k} E(\theta)\big|_{\theta=\theta^{(n)}} \tag{9}$$

using a small learning rate $\tau$.

However, the energy landscape $E(\theta)$ is typically not convex, meaning that the plain gradient descent optimization is prone to getting stuck in saddle points or local minima. Among other choices of advanced optimizers, the convergence can be accelerated by employing the Stochastic Reconfiguration (SR) method [42], where the update step (9) is altered to

$$\theta_k^{(n+1)} = \theta_k^{(n)} - \tau \mathcal{S}_{k,k'}^{-1} \partial_{\theta_{k'}} E(\theta)\big|_{\theta=\theta^{(n)}} \ . \tag{10}$$

In this expression $\mathcal{S}_{k,k'} = \mathrm{Re}\left(S_{k,k'}\right)$ is the real part of the quantum Fisher matrix

$$S_{k,k'} = \left\langle\!\!\left\langle \left(\mathcal{O}_k^\theta\right)^* \mathcal{O}_{k'}^\theta \right\rangle\!\!\right\rangle_\theta^c \ . \tag{11}$$

The quantum Fisher matrix is the metric tensor of the Fubini-Study metric – a natural metric on the projective Hilbert space [43]. It provides information about the local geometry of the variational manifold, which is exploited by adjusting the gradient step accordingly in Eq. (10). However, this approach requires the inversion of the $\mathcal{S}$-matrix, which imposes constraints on the feasible number of variational parameters that determines the matrix dimension. Moreover, the quantum Fisher matrix is often (approximately) rank-deficient, which means that changing the parameters along some directions in the variational manifold leaves the physical state invariant. This ill-conditionedness demands careful regularization when inverting $\mathcal{S}$ for the SR update step in Eq. (10), see Section 2.5 for details.

Besides ground states, it is also possible to address low-lying excited states with VMC techniques. In the method introduced in Ref. [44] the first step is to perform a ground state search resulting in a set of optimal parameters $\theta^*$ and a variational ground state $|\varphi(\theta^*)\rangle$. Subsequently, the ansatz for the excited state is projected onto the subspace orthogonal to the ground state by defining

$$|\Psi_{\mathrm{exc.}}(\theta)\rangle \equiv |\psi(\theta)\rangle - \lambda|\varphi(\theta^*)\rangle \tag{12}$$

with

$$\lambda = \frac{\langle \varphi(\theta^*)|\psi(\theta)\rangle}{\langle \varphi(\theta^*)|\varphi(\theta^*)\rangle} = \sum_{\mathbf{s}} \frac{|\varphi_{\theta^*}(\mathbf{s})|^2}{\langle \varphi(\theta^*)|\varphi(\theta^*)\rangle} \frac{\psi_\theta(\mathbf{s})}{\varphi_{\theta^*}(\mathbf{s})} = \left\langle\!\!\left\langle \frac{\psi_\theta}{\varphi_{\theta^*}} \right\rangle\!\!\right\rangle_{\theta^*} . \tag{13}$$

The excited state search is then performed with alternating steps, first computing $\lambda$ and then performing one SR step with the ansatz (12).

## 2.3 Unitary dynamics

For the simulation of real time dynamics the goal is to obtain equations of motion for the time-dependent variational parameters $\theta(t)$ that yield an approximate solution of the Schrödinger equation

$$i\frac{d}{dt}|\psi(\theta(t))\rangle = \hat{H}|\psi(\theta(t))\rangle \ , \tag{14}$$

where $\hat{H}$ denotes the system Hamiltonian. While alternative approaches have been explored recently [22], there are two long established ways to formulate a TDVP for the Schrödinger equation – (i) based on the principle of least action [45,46], and (ii) based on the maximization of overlaps or similar measures of proximity in the underlying Hilbert space [10,46,47]. These two derivations yield the identical result in the special case, where the parametrized wave function $\psi_\theta(\mathbf{s}) \equiv \psi_\vartheta(\mathbf{s})$ is holomorphic, i.e., fulfills Cauchy-Riemann equations as function of complex variational parameters $\vartheta_l = \theta_{2l} + i\theta_{2l+1} \in \mathbb{C}$ [46]. The corresponding TDVP equation is a first order differential equation for the variational parameters:

$$S_{l,l'}\dot{\vartheta}_{l'} = -iF_l \tag{15}$$

Here, the force vector is defined as

$$F_l = \left\langle\!\!\left\langle E_{\mathrm{loc}}^\vartheta \left(\mathcal{O}_l^\vartheta\right)^* \right\rangle\!\!\right\rangle_\vartheta^c \ , \tag{16}$$

and $S_{l,l'}$ is the quantum Fisher matrix as in Eq. (11). In this case $\mathcal{O}_l^\vartheta(\mathbf{s}) = \partial_{\vartheta_l} \log \psi_\vartheta(\mathbf{s})$ denotes the complex derivative of the logarithmic wave function coefficients.

The TDVP equation (15) defines a Hamiltonian dynamics of the variational parameters, which conserves the quantum expectation value of energy [45]. Notice, however, that other constants of motion of the quantum dynamics are in general not conserved under this TDVP.

When releasing the requirement of a holomorphic ansatz, the different approaches to the TDVP yield closely related, but different results. The principle of least action, (i), results in

$$\mathrm{Im}\left[S_{k,k'}\right]\dot{\theta}_{k'} = \mathrm{Im}\left[-iF_k\right] \ , \tag{17}$$

while the minimization of some distance measure, (ii), gives

$$\mathrm{Re}\left[S_{k,k'}\right]\dot{\theta}_{k'} = \mathrm{Re}\left[-iF_k\right] . \tag{18}$$

While both equations correspond to a valid TDVP, only Eq. (17) preserves the symplectic structure of the variational manifold and thereby the conservation of energy under time evolution; by contrast, following Eq. (18), energy is not necessarily conserved. Moreover, solving Eq. (15) for complex parameters $\vartheta_l = \theta_{2l} + i\theta_{2l+1}$ is equivalent to solving Eq. (17) for the corresponding real parameters $\theta_k$.

For a unified notation we can write

$$[[S_{k,k'}]]\dot{\theta}_{k'} = -[[iF_k]] \, , \tag{19}$$

where $[[\cdot]]$ denotes the identity, $\text{Re}[\cdot]$ or $\text{Im}[\cdot]$, depending on which version of the TDVP is used. Accordingly, inverting $[[S]]$ yields an ordinary differential equation in the standard form, which can be integrated in order to obtain the time-evolved wave function. Again, $[[S]]$ is typically ill-conditioned, meaning that a suited regularization has to be applied in order to obtain a stable solution, see Section 2.5.

As an aside, notice that for the purpose of VMC we consider wave functions $\psi_\theta(\mathbf{s})$ with no further constraints on $\theta$. Therefore, the solution of Eq. (19) is always part of the variational manifold we consider. A prominent example of an ansatz class with additional constraints on the variational parameters $\theta$ are the matrix product states. In that case an explicit projection onto the variational manifold has to be included as part of the TDVP [47, 48].

Stochastic Reconfiguration for ground state search and the TDVP for unitary time evolution are closely related: Since $\partial_{\theta_k} E(\theta) = 2\text{Re}[F_k]$, the SR update step (10) corresponds to an *imaginary* time step according to the TDVP equation. Therefore, the computational steps required for both are very similar and they are summarized in the pseudocode of Algorithm 1. We will see in the following section that also variational simulation of the dynamics of an open quantum system shares the same building blocks. The jVMC codebase provides efficient implementations of these fundamental ingredients for NQS algorithms.

---

**Algorithm 1** Single SR or TDVP time step

---

1: **procedure** TIMESTEP(psi, H, n_Samples)
2:     configs ← psi.sample(n_Samples)                          ▷ obtain samples
3:     amp_Configs ← psi.evaluate(configs)               ▷ obtain sample amplitudes
4:     gradients ← psi.gradients(configs)     ▷ obtain gradients of the sample amplitudes
5:
6:     offdConfigs, offdElements ← H.get_offdConfigs(configs)   ▷ obtain coupled configs
7:     amp_offdConfigs ← psi.evaluate(offdConfigs)       ▷ evaluation on coupled configs
8:     $E_{\text{loc}}$ ← H.get_O_loc(amp_Configs, amp_offdConfigs)       ▷ compute local energies
9:
10:     S, F ← get_TDVP_equation(gradients, $E_{\text{loc}}$)
11:     $\dot{\theta}$ ← solve_TDVP_equation(S, F)                          ▷ solve TDVP equation
12:
13:     **return** $\dot{\theta}$

---

## 2.4 Markovian dissipative dynamics

When taking into account the coupling to an environment, quantum states become mixed and one has to resort to the density matrix formalism for a theoretical description. Assuming a Markovian environment, the time evolution of the density matrix is given by the Lindbladian extension of the von-Neumann equation

$$\frac{d}{dt}\hat{\rho} = -i[\hat{H}, \hat{\rho}] + \sum_i \left[ \hat{L}^i \hat{\rho} \hat{L}^{i\dagger} - \frac{1}{2} \left\{ \hat{L}^{i\dagger} \hat{L}^i, \hat{\rho} \right\} \right] \, . \tag{20}$$

Here, $[\cdot, \cdot]$ and $\{\cdot, \cdot\}$ denote the commutator and the anti-commutator, respectively. Moreover the operators $\hat{L}^i$ are the so-called jump operators, which incorporate the action of the environment and drive the system towards a steady-state.

The TDVP for unitary dynamics can be generalized to variational approaches to solve the Lindblad equation. One possibility is based on representing the density matrix in a purified form [27]. While implementing the purification approach is also within the scope of the jVMC codebase, we focus in the following on an alternative method that relies on the Positive Operator Valued Measurements (POVM)-formalism [28, 29, 34, 49, 50] for a purely probabilistic formulation of quantum mechanics.

A POVM is a set of positive semi-definite operators $\{\hat{M}^a\}$ such that $\sum_a \hat{M}^a = \mathbb{1}$. Given a POVM $\{\hat{M}^a\}$ for the local Hilbert space, a many-body POVM can be constructed as $\hat{M}^{\mathbf{a}} = \hat{M}^{a_1} \otimes \ldots \otimes \hat{M}^{a_N}$ [34] and the outcome $\mathbf{a} = (a_1, \ldots, a_N)$ is associated with the probability

$$P^{\mathbf{a}} = \operatorname{tr}\left(\hat{\rho}\hat{M}^{\mathbf{a}}\right) . \tag{21}$$

A POVM is called minimal and informationally complete if the number of POVM elements equals the number of free parameters of the density matrix. In that case the POVM forms an operator basis on the considered Hilbert space and the relation in Eq. (21) is invertible such that

$$\hat{\rho} = \sum_{\mathbf{a},\mathbf{b}} \hat{M}^{\mathbf{a}}\left(T^{-1}\right)^{\mathbf{ab}} P^{\mathbf{b}} \tag{22}$$

with the overlap matrix $T^{\mathbf{ab}} = \operatorname{tr}\left(\hat{M}^{\mathbf{a}}\hat{M}^{\mathbf{b}}\right)$. Hence, a minimal informationally complete POVM yields an equivalent purely probabilistic formulation of quantum mechanics in which the dissipative dynamics is governed by a Master equation for the probability distribution $P^{\mathbf{a}}$,

$$\dot{P}^{\mathbf{a}} = \mathcal{L}^{\mathbf{ab}} P^{\mathbf{b}} . \tag{23}$$

In this expression, the linear operator $\mathcal{L}^{\mathbf{ab}}$ contains the physical information equivalent to the Lindblad equation (20), see Ref. [29] for details.

For the variational approach, an ansatz $P_\theta^{\mathbf{a}}$ is chosen in order to find an efficient representation of the POVM distribution, and a TDVP yields a first-order differential equation for the variational parameters $\theta$,

$$S_{k,k'}\dot{\theta}_{k'} = F_k , \tag{24}$$

see Ref. [29]. Here, $S$ denotes the Fisher matrix $S_{k,k'} = \langle\!\langle \mathcal{O}_k^{\mathbf{a}} \mathcal{O}_{k'}^{\mathbf{a}} \rangle\!\rangle_{\mathbf{a}\sim P}^c$, and $F_k = \langle\!\langle \mathcal{O}_k^{\mathbf{a}} \mathcal{L}^{\mathbf{ab}} \frac{P^{\mathbf{b}}}{P^{\mathbf{a}}} \rangle\!\rangle_{\mathbf{a}\sim P}^c$, where the repeated indices $\mathbf{b}$ inside the brackets are summed over. The double brackets denote connected correlation functions $\langle\!\langle AB \rangle\!\rangle^c = \langle\!\langle AB \rangle\!\rangle - \langle\!\langle A \rangle\!\rangle\langle\!\langle B \rangle\!\rangle$ of expectation values with respect to the POVM-distribution $P$ and $\mathcal{O}_k^{\mathbf{a}} = \partial_{\theta_k} \log P^{\mathbf{a}}$.

## 2.5 Regularization to invert the (quantum) Fisher matrix

In many cases the (quantum) Fisher matrix turns out the be ill-conditioned with an eigenvalue spectrum that spans all numerical orders of magnitude. Therefore, ground state search with SR and time evolution via TDVP require suited regularization schemes when inverting the Fisher matrix. Three possible approaches are outlined below. While either the diagonal shift (Section 2.5.1) or the pseudo-inverse regularization (Section 2.5.2) should typically be used for SR, the effective regularization for real time evolution, e.g., targeted elimination of noisy contributions (Section 2.5.3), is under ongoing investigation and there are no generally established procedures to date [24, 51].

### 2.5.1 Diagonal shift

An effective regularization for SR is to introduce a diagonal shift, i.e., to replace the Fisher matrix $S_{k,k'}$ by

$$\tilde{S}_{k,k'} = \big(1 + \nu \delta_{k,k'}\big) S_{k,k'} \; . \tag{25}$$

Artificially amplifying the diagonal elements of the $S$-matrix mitigates the rank-deficiency and renders the matrix invertible. The shift parameter $\nu$ is typically reduced during the optimization according to a suited schedule [10].

The regularization via diagonal shift is not applicable for time evolution, because the TDVP relies on an accurate solution of the actual TDVP equation at each time point; by contrast, the ground state search is more forgiving in this regard because one is only interested in the final solution and typically not in the trajectory during the optimization. Moreover, other than the regularization schemes discussed below, the diagonal shift regularization can be combined with iterative solvers for the linear equation $S_{k,k'}\dot{\theta}_{k'} = F_k$, such as conjugate gradient [10].

### 2.5.2 Pseudo-inverse

The Moore-Penrose pseudo-inverse generalizes matrix inversion to rank-deficient matrices. The pseudo-inverse $S^+$ of a hermitian/symmetric matrix $S$ with eigendecomposition $S = VDV^\dagger$, where $D = \mathrm{diag}(\lambda_1, \ldots, \lambda_K)$, can be constructed as $S^+ = V\mathrm{diag}(\lambda_1^+, \ldots, \lambda_K^+)V^\dagger$, where

$$\lambda_j^+ = \begin{cases} 0 & \text{if } \lambda_j = 0 \\ \lambda_j^{-1} & \text{if } \lambda_j \neq 0 \end{cases} \tag{26}$$

For the case of ill-conditioned matrices with small but non-vanishing eigenvalues this can be generalized to an approximate pseudo-inverse with

$$\lambda_j^+ = \begin{cases} 0 & \text{if } |\lambda_j/\lambda_1| < \epsilon_{\mathrm{pinv}} \\ \lambda_j^{-1} & \text{if } |\lambda_j/\lambda_1| \geq \epsilon_{\mathrm{pinv}} \end{cases} \tag{27}$$

where we introduced the cutoff parameter $\epsilon_{\mathrm{pinv}}$ and assumed ordered eigenvalues $|\lambda_1| \geq \ldots \geq |\lambda_K|$.

For real time evolution it can be beneficial to choose a soft cutoff instead of Eq. (27), because eigenvalues that cross the cutoff can otherwise introduce spurious discontinuities in the dynamics. One possibility is

$$\lambda_j^+ = \left[ \lambda_j \left( 1 + \Big( \frac{\epsilon_{\mathrm{pinv}}}{|\lambda_j/\lambda_1|} \Big)^6 \right) \right]^{-1} \; . \tag{28}$$

The regularization with approximate pseudo-inverses requires diagonalization of the (quantum) Fisher matrix. Since the size of the Fisher matrix is determined by the number of variational parameters, this imposes immediate restrictions on the size of the variational ansatz.

### 2.5.3 Eliminating noisy contributions

Monte Carlo fluctuations that are blown up by the inversion of small eigenvalues of the Fisher matrix constitute a major source of instabilities when simulating real time evolution [24,51]. A mitigation strategy introduced in Ref. [24] takes the signal-to-noise ratio (SNR)

of the MC estimates in the TDVP equation into account for the regularization. Denoting the force vector $F_k$ transformed to the eigenbasis of $S$ as $\rho_k = V_{k,k'}^\dagger F_{k'}$, its signal-to-noise ratio $\mathrm{SNR}(\rho_k)$ can be estimated using the available MC data. Then, a regularization that ignores exceedingly noisy components of the TDVP equation can be defined in analogy to the pseudo-inverse with

$$\lambda_j^+ = \left[ \lambda_j \left( 1 + \left( \frac{\epsilon_{\mathrm{SNR}}}{\mathrm{SNR}(\rho_k)} \right)^6 \right) \right]^{-1} . \tag{29}$$

The cutoff parameter defines a lower bound for the $\epsilon_{\mathrm{SNR}}$ that is tolerated and the number of discarded components can be tuned systematically by varying the number of MC samples, and thereby the SNR.

This SNR-based regularization can be combined with the pseudo-inverse approach discussed above.

## 2.6 Neural quantum states

In the discussion of the previous sections we made no reference to any specific choice of the variational ansatz $\psi_\theta(\mathbf{s})$ (or $P_\theta^{\mathbf{a}}$). While these variational approaches are formulated for arbitrary wave function ansatzes, the particular choice will in general introduce a bias to the obtained results. Numerically exact simulations, however, require the possibility to systematically reduce this bias.

Artificial neural networks (ANNs) have been proven to be universal function approximators in the limit of large network sizes [52–56]. Hence, by choosing ANNs as the variational ansatz for the wave function the inductive bias of the NQS can be reduced systematically by increasing the network size. Thereby, the network size plays a similar role for NQS as the bond dimension in the established tensor network techniques [2–4].

In its most basic form, the ANN is a function that is composed from an alternating sequence of affine-linear and non-linear transformations. Each affine-linear transformation maps a vector of activations $\mathbf{a}^{(l-1)}$ to a new vector of pre-activations

$$z_i^{(l)} = \sum_j W_{ij}^{(l)} a_j^{(l-1)} + b_i^{(l)} , \tag{30}$$

from which the new activations are obtained by applying a non-linear function $\sigma(\cdot)$ element-wise,

$$a_i^{(l)} = \sigma\left( z_i^{(l)} \right) . \tag{31}$$

This iterative prescription is initialized by identifying $\mathbf{a}^{(0)}$ with the input of the ANN. The set of weights $W_{ij}^{(l)}$ and biases $b_i^{(l)}$ constitutes the variational parameters, which we will in the following summarize in the variable $\theta$.

For our purposes, we assume that the ANN encodes the logarithm of the wave function coefficients, i.e.,

$$\log \psi_\theta(\mathbf{s}) \equiv f_\theta(\mathbf{s}) \tag{32}$$

where $f_\theta(s)$ represents the ANN. In the following two subsections we discuss two further central design choices, namely holomorphic vs. non-holomorphic ANNs and the possibility to equip the NQS with an autoregressive structure to enable direct sampling. On this basis, arbitrary ANN architectures can be incorporated in the NQS, for example, Restricted Boltzmann Machines [10], convolutional [15], recurrent [57], sparse [58], or graph neural networks [59].

### 2.6.1 (Non-)holomorphic neural quantum states

The wave function coefficients $\psi_\theta(\mathbf{s})$ are in general complex numbers, which can be obtained in different ways from an ANN. Three possible options are:

1. Single holomorphic network: Allow the weights and biases to be complex numbers, $\vartheta \in \mathbb{C}$, and choose holomorphic activation functions $\sigma$. Then $f_\vartheta$ itself is a holomorphic function of the variational parameters $\vartheta$.

2. Single non-holomorphic network: Construct an ANN with real parameters, $\theta \in \mathbb{R}$, with two real outputs, $f_\theta(\mathbf{s}) = \left( f_\theta^{(1)}(\mathbf{s}), f_\theta^{(2)}(\mathbf{s}) \right) \in \mathbb{R}^2$. This yields a complex wave function coefficient through

$$\log \psi_\theta(\mathbf{s}) = f_\theta^{(1)}(\mathbf{s}) + i f_\theta^{(2)}(\mathbf{s}) \ . \tag{33}$$

3. Two real (non-holomorphic) networks: Use two separate ANNs $f_{\theta^{(f)}}$ and $g_{\theta^{(g)}}$ for real and imaginary part

$$\log \psi_\theta(\mathbf{s}) = f_{\theta^{(f)}}(\mathbf{s}) + i g_{\theta^{(g)}}(\mathbf{s}) \ . \tag{34}$$

In this case, independent variational parameters are used for the real and the imaginary part, $\theta = \left( \theta^{(f)}, \theta^{(g)} \right)$.

### 2.6.2 Autoregressive neural quantum states

Any joint probability distribution $p(\mathbf{s}) = p(s_1, \ldots, s_N)$ can be factorized and written as a product of conditional probabilities:

$$p(s_1, \ldots, s_N) = p_1(s_1) p_2(s_2|s_1) p_3(s_3|s_1, s_2) \ldots p_N(s_N|s_1, \ldots, s_{N-1}) \ . \tag{35}$$

If the random variables $s_i$ have few discrete outcomes, this property can be exploited to efficiently generate uncorrelated samples $\mathbf{s} \sim p$ without resorting to Markov Chain Monte Carlo. Instead, the individual realizations of $s_i$ are sampled directly in a sequential manner. Since the $s_i$ have only few outcomes, it is straightforward to draw a sample outcome for $s_1$ from $p_1$. Subsequently, the same is true for $s_j$ with $j > 1$, because $p_j(s_j|s_1, \ldots, s_{j-1})$ is again a distribution with few outcomes for the given $s_1, \ldots, s_{j-1}$.

ANNs, which incorporate this factorization into conditionals are called autoregressive models. There are different architectures with this property, such as Neural Autoregressive Density Estimators [11,60] or recurrent neural networks [57]. Utilizing direct sampling can be advantageous in cases where Markov Chain Monte Carlo (MCMC) becomes inefficient due to exceedingly long autocorrelation times [11]. In fact, direct sampling is also more efficient than MCMC when autocorrelation times are short. In MCMC consecutive samples are collected after performing one update sweep that typically consists of one Monte Carlo step per degree of freedom. Hence, generating one sample with MCMC comes at the cost of $\mathcal{O}(N)$ network evaluations, where $N$ is the system size. By contrast, generating one sample costs just one network evaluation when exploiting the autoregressive property.

However, the autoregressive property imposes specific constraints on the network architecture. Therefore, the advantage of more efficient sampling has to be weighed against inductive biases of the architecture.

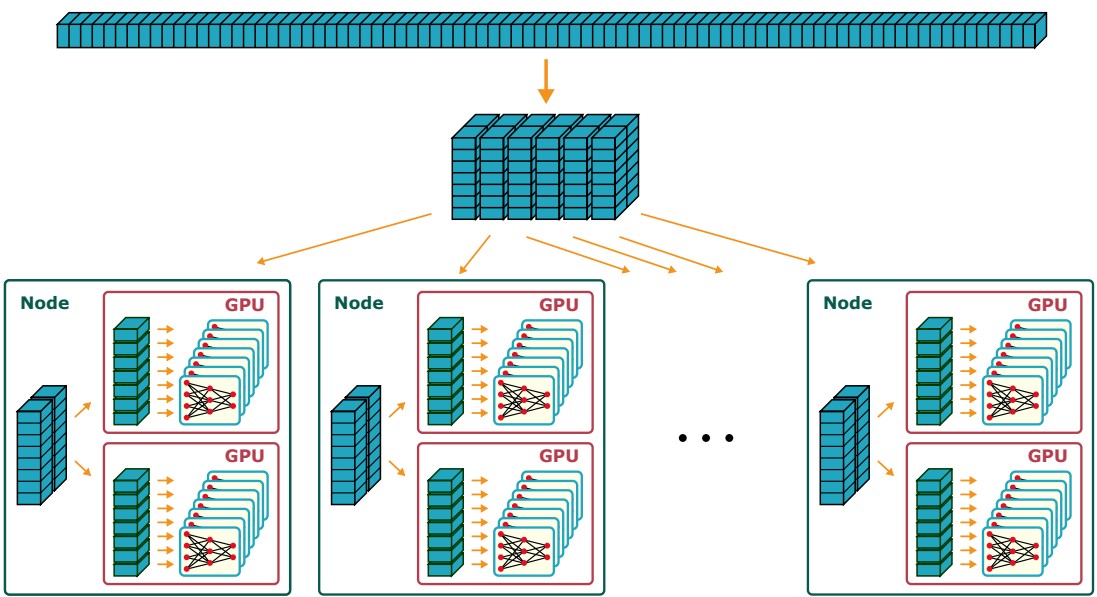

Figure 1: Schematic depiction of the parallelization scheme. Each of the blue boxes represents the task to evaluate the NQS on an individual basis configuration. These tasks are evenly distributed across the available compute nodes and separated into batches, which can be executed concurrently by vectorized operations on the local accelerators.

# 3 Design choices

The jVMC codebase is devised as a transparent implementation of the core computational tasks required when dealing with NQS, which simultaneously provides efficiency and large flexibility. The primary purpose is to equip the user with these building blocks that enable the composition of a large variety of algorithms.

The code is designed to fully exploit the algorithms' typical amenability to hybrid parallelization, combining distributed single program multiple data (SPMD) computing with vectorization that can benefit substantially from the availability of accelerators like GPUs. Automatic differentiation is used to enable the composition of arbitrary ansatz functions for the variational quantum states. The code relies on the JAX library [36], which provides the functionality for automatic differentiation as well as vectorization and just-in-time compilation; see Appendix A for basic examples illustrating JAX's functionality.

In this section we first explain how different levels of parallelism are exploited and how this is reflected in the API before describing the core modules of the package.

## 3.1 Parallelism

The computationally intense part of NQS-based algorithms is typically the evaluation of the ANN on large numbers of computational basis states. Since each individual ANN evaluation is independent of the others and largely consists of vectorizable operations, these algorithms are well suited to exploit the resources of distributed compute clusters with accelerators. The parallelization scheme that is supported by the jVMC codebase is depicted in Fig. 1. In the depiction the small blue boxes represent the task to evaluate the ANN on an individual computational basis configuration. The total set of input configurations is distributed across the available compute nodes, where, in turn, the assigned configurations are split up among the locally available accelerators. The accelerators work

most efficiently when evaluating the ANN on large batches of input configurations simultaneously, see Section 3.1.2 below.

This parallelization scheme applies straightforwardly to the computation of observables following Eq. (4) when a large number of sample configurations $\mathbf{s}$ is given and it is applicable in a similar fashion to MC sampling. To achieve optimal performance in MCMC, it is beneficial to assign multiple independent Monte Carlo chains to each accelerator. The MCMC steps of all chains on one device can be performed in sync, again allowing for vectorized ANN evaluation on mutliple input configurations.

The code was designed with this hybrid parallelization scheme as a guiding principle. An important manifestation are the required array dimensions when interfacing jVMC: All data that is related to network evaluations will have two leading dimensions, namely the *device dimension* and the *batch dimension*. These are indicated by the stacks, which are assigned to individual nodes in Fig. 1. The corresponding layers of parallelism, multiple accelerators per node and vectorization, are explained in more detail in the following subsections.

### 3.1.1 Multiple accelerators per node

The jVMC codebase supports two possibilities to deal with compute clusters where multiple accelerators are attached to each node, namely,

1. Launch one MPI process per accelerator.

2. Distribute computation across the available devices, while working with a single process.

While the former is straightforward using the `mpi4py` package, the latter is enabled by automatic parallelization across devices using the `pmap` functionality of the JAX library. The `jVMC.set_pmap_devices()` function enables the user at the beginning of a program to choose for each MPI process which subset of the available devices to work with. For a homogeneous treatment of both options, all data arrays that are passed through the jVMC API have an additional leading *device dimension* to account for potential parallelization across devices. The size of this dimension corresponds to the number of devices used by the process and any computation will be distributed among the devices.

It is important to realize and keep in mind that when working with multiple devices the device dimension is also physically distributed across the different devices. Hence, any computation on data with device dimension larger than one should be performed on the respective devices to avoid memory transfer overheads.

The default behavior of the JAX library is to allocate all available GPU memory at the beginning of a program. This can lead to conflicts if multiple processes can access the same GPU devices. This issue can be avoided by setting the `XLA_PYTHON_CLIENT_PREALLOCATE` environment variable prior to running the program as

```
>>> export XLA_PYTHON_CLIENT_PREALLOCATE=false
```

### 3.1.2 Batching for vectorization

The computationally intense operation during network evaluations is the matrix-vector product of the affine-linear transformation in Eq. (30), which can be turned into a matrix-matrix product when evaluating the ANN simultaneously on a batch of input configurations. The arithmetic intensity of multiplying a $m \times n$-matrix with a $n \times k$-matrix, i.e., the

number of arithmetic operations per memory access, is

$$I(m, n, k) = \frac{mnk}{mn + nk + mk} \ .\tag{36}$$

This contains the arithmetic intensity of a matrix-vector product as the special case with $k = 1$, which is bounded from above irrespective of the values of $m$ and $n$:

$$I(m, n, 1) = \frac{mn}{mn + n + m} = \frac{1}{1 + \frac{n}{m} + \frac{m}{n}} < 1\tag{37}$$

By contrast, $I(m, n, k)$ grows with increasing $k$ (linearly for $k \ll \min(m, n)$), meaning for the typical affine-linear transformation of ANNs that a suited batching of computational tasks can substantially enhance the arithmetic intensity. Therefore, any operation implemented in jVMC is performed on a batch of input data. This means that following the leading device dimension, all data arrays have an additional *batch dimension*, which should always be chosen as large as possible given the available memory in order to keep the arithmetic units of the GPU busy.

## 3.2 Core modules

In the following we describe the main functionality of the core modules of the codebase. A detailed documentation is available online [40].

Fig. 2 provides an overview of the main modules. The core functionality is contained in `jVMC.vqs`, `jVMC.sampler`, and `jVMC.operator`. These implement the central operations of typical NQS algorithms in an efficient and flexible manner. The `jVMC.util` and the `jVMC.mpi_wrapper` modules provide higher-level functionality useful for typical NQS algorithms and functionality for the evaluation of distributed Monte Carlo samples, respectively. The main features of these five modules are described in the following.

A recurring pattern is the possibility to create custom objects, e.g., network architectures or quantum operators. Their behavior has to be defined in the form of functions acting on a computational basis configuration. As a consistent design choice, these functions are always defined acting on a *single* input configuration. The vectorization for batched evaluations is then taken care of automatically within the respective classes of the jVMC codebase.

### 3.2.1 Variational wave functions

A core part of the jVMC codebase is the `NQS` class provided in the `jVMC.vqs` module – a wrapper class for variational wave functions, which provides an interface that other parts of the code rely on. The `NQS` class is used to wrap ANNs that are defined using the Flax module system Linen [61]. In this framework ANNs are defined in a very similar manner as in the popular Pytorch or Keras packages as classes that are derived from the `flax.linen.Module` base class. The minimal requirement for this new class is the implementation of a `__call__` member function, which evaluates the neural network on a single input configuration. For example, a Restricted Boltzmann Machine (RBM) with complex parameters, for which

$$\log \psi_\theta(\mathbf{s}) = \sum_{i=1}^{N} a_i s_i + \sum_{i=1}^{N_h} \log \left( \cosh \left( \sum_{j=1}^{N} W_{ij} s_j + b_j \right) \right)\tag{38}$$

with $N_h$ the number of hidden units, can be defined as follows:

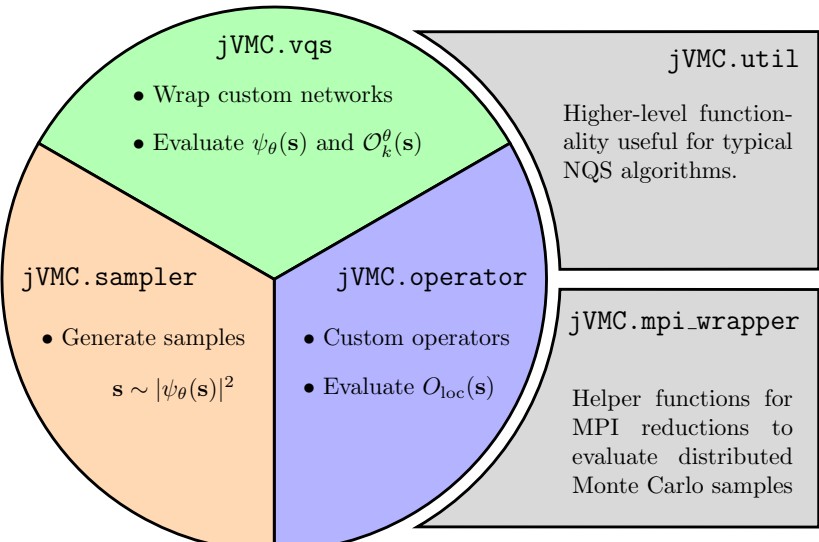

Figure 2: Overview of the `jVMC` codebase. The core modules (colored) implement the central operations of typical NQS algorithms in an efficient and flexible manner.

```python
class MyRBM(flax.linen.Module):
    numHidden: int = 2 # number of hidden units

    @flax.linen.compact
    def __call__(self, s):

        s = 2 * s - 1 # Go from 0/1 representation to 1/-1

        # Apply dense layer
        h = flax.linen.Dense(features=self.numHidden,
                             dtype=jVMC.global_defs.tCpx)(s)

        # Apply activation function
        h = jax.numpy.log(jax.numpy.cosh(h))

        # Visible bias
        vbias = self.param("vbias", jax.nn.initializers.zeros, s.shape)

        return jax.numpy.sum(h) + jax.numpy.dot(vbias, s)
```

Notice in this example code that the `flax.linen` package provides basic modules like, e.g., dense layers (`flax.linen.Dense`), that implement the affine transformation in Eq. (38). Moreover, the `flax.linen.compact` decorator enables the inlined definition of such submodules, which otherwise have to be defined in a separate `setup` member function (see Flax documentation for details [61]).

In order to work with autoregressive NQS (see Section 2.6.2), a member function called `sample` has to be implemented as part of the corresponding Flax Linen module. This `sample` member function has to take the desired number of samples and a `jax.random.PRNGKey` as input arguments and it should return the resulting configurations.

The `jVMC.nets` module contains a number of pre-defined common network architectures, including Restricted Boltzmann Machines, Convolutional Neural Networks, and

Recurrent Neural Networks. Notice, however, that these classes of architectures delineate general design principles, but leave a lot of freedom regarding the details of the implementation. The jVMC codebase is intended to encourage the exploration of new architectures, for which the provided examples should be regarded as possible starting points.

Once the network architecture is defined in the form of a `flax.linen.Module`, an instance of it can be initialized. Continuing the example from above, an RBM can be obtained as follows:

```
>>> net = MyRBM(numHidden=7) # Initialize custom ANN
```

The `NQS` class supports the three types of NQS introduced in Section 2.6.1, i.e., a single holomorphic or non-holomorphic ANN or two separate ANNs encoding logarithmic amplitude and phase, respectively. The corresponding ANNs in the form of Linen modules are passed to the `NQS` class at instantiation. In our example, we define a holomorphic RBM NQS with the following line:

```
>>> psi = jVMC.vqs.NQS(net) # Initialize an NQS object
```

In order to use two separate networks for phase and amplitude, the two networks are passed as a tuple. In addition to this, the constructor takes as optional arguments a `batchSize` (the meaning of which is explained below) and a `seed` for the initialization of the network parameters.

The main purpose of the `NQS` class is to provide an interface for computing wave function coefficients as well as gradients of the variational ansatz. For a given sample of input configurations $\mathbf{s}$ the `__call__` method yields the logarithmic wave function coefficients $\log\psi_\theta(\mathbf{s})$ and the `gradients` method returns the logarithmic gradients $\mathcal{O}_k^\theta(\mathbf{s}) = \partial_{\theta_k}\log\psi_\theta(\mathbf{s})$. These methods automatically vectorize the evaluation, meaning that the input requires the additional device and batch dimension as explained in Section 3.1 above. Since the batch-vectorization is limited by the available memory, the batch is prior to evaluation split into mini-batches of size `batchSize`, which are evaluated simultaneously. The `batchSize` used by the `NQS` class is fixed at instantiation (see above) and it should be chosen as large as possible with the given memory resources in order to render the network evaluation compute-bound, see Section 3.1.2.

Finally, the `NQS` class provides member functions `get_parameters`, `set_parameters`, and `update_parameters` to manipulate the network parameters; see the documentation for details [39].

### 3.2.2 Operators

The `jVMC.operator` module implements the evaluation of the action of operators on computational basis configurations. In terms of an operator $\hat{O}$ acting on the Hilbert space this corresponds to "on-the-fly" generation of matrix elements. Generally, this amounts to a mapping

$$\hat{O} : \mathbf{s} \mapsto \{\mathbf{s}'_j\}, \{O_{\mathbf{s}\mathbf{s}'_j}\} \tag{39}$$

Here, $\mathbf{s}$ denotes the input basis configuration and the $\mathbf{s}'_j$ are the basis configurations for which the matrix elements are non-zero, $O_{\mathbf{s}\mathbf{s}'_j} = \langle\mathbf{s}|\hat{O}|\mathbf{s}'_j\rangle \neq 0$.

The abstract `Operator` class defines an interface for the implementation of mappings given in Eq. (39). Any specific operator can be implemented as a child of `Operator`, which has to implement a `compile()` member function. Such an implementation should return a JAX-*jit-able* function (see Appendix A.1) that returns for a given basis configuration $\mathbf{s}$

a tuple of two arrays, one containing the configurations $\mathbf{s}'_j$ – the other the corresponding matrix elements $O_{\mathbf{ss}'_j}$.

The following example code shows the implementation of a $\hat{\sigma}_l^x$ operator acting on a chain of spin-1/2 degrees of freedom, i.e., $\mathbf{s} \in \{0, 1\}^N$:

```python
class SxOperator(jVMC.operator.Operator):
    """Define a '\hat\sigma_l^x' operator."""

    def __init__(self, siteIdx):

        self.siteIdx = siteIdx

        super().__init__() # Constructor of base class Operator has to be called!

    def compile(self):

        def get_s_primes(s, idx):

            # Create copy of input
            sp = s.copy()

            # Define matrix element
            matEl = jax.numpy.array([1., ], dtype=global_defs.tCpx)
            # Define mapping of Sx: 0->1, 1->0
            sMap = jax.numpy.array([1, 0])
            # Perform mapping
            sp = jax.ops.index_update(sp, jax.ops.index[idx], sMap[s[idx]])

            return sp, matEl

        # Create a pure function that takes only a basis configuration as argument
        map_function = functools.partial(get_s_primes, idx=self.siteIdx)

        return map_function
```

As mentioned above, the new `SxOperator` class inherits from the abstract `Operator` class. Crucially, the parent constructor has to be called at the end of the constructor of the new class, i.e., `super().__init__()` has to be included. The `compile` member function constructs and returns a function that defines the operator's action. In our example we can test its behavior as follows:

```python
>>> mySx = SxOperator(siteIdx=1) # Initialize operator
>>> testFunction = mySx.compile() # Get operator evaluation function
>>> testConfig = jax.numpy.array([0, 0, 0, 0], dtype=np.int32) # Test
    configuration
>>> sp, matEl = testFunction(testConfig) # Evaluate operator on test
    configuration
>>> print(sp)
[0 1 0 0]
>>> print(matEl)
[1.+0.j]
```

Any child of the abstract `Operator` class inherits the following member functions, which employ the operator action defined by the specific `compile` function:

- `get_s_primes(s)`: Returns $\mathbf{s}'_j$ and $O_{\mathbf{ss}'_j}$ for a batch of input states $\mathbf{s}$. In accordance with Section 3.1 the input dimension is $D \times B \times$ (spatial dimensions), where $D$ is

the device dimension and $B$ is the batch size. The output is a tuple of two arrays of shape $D \times M \times$ (spatial dimensions) and $D \times M$, where $M$ is the total number of non-zero matrix elements across the whole batch.

- `get_O_loc(logPsiS, logPsiSP)`: Assuming that `get_s_primes(s)` has been called before on a batch of input states $\mathbf{s}$, this function computes the corresponding $O_{\text{loc}}^\theta(\mathbf{s})$ for each element of the batch. As input it requires the logarithmic wave function coefficients $\log \psi_\theta(\mathbf{s})$ (argument `logPsiS`) and $\log \psi_\theta(\mathbf{s}')$ (argument `logPsiSP`). It returns an array of size $D \times B$ with the complex-valued $O_{\text{loc}}^\theta(\mathbf{s})$.

Thereby, children of the `Operator` class provide comprehensive functionality to evaluate $O_{\text{loc}}^\theta(\mathbf{s})$ for arbitrary operators $\hat{O}$ in an automatically vectorized manner. Consider, for example, that `H` is an operator object implementing the action of a Hamiltonian $\hat{H}$, `psi` is the variational wave function, i.e., an instance of the `NQS` class, and `sampleConfigs` is a batch of basis configurations. Then, evaluating $E_{\text{loc}}^\theta(\mathbf{s})$ is achieved by the following lines of code:

```
sampleOffdConfigs, matEls = H.get_s_primes(sampleConfigs)
sampleLogPsi = psi(sampleConfigs)
sampleLogPsiOffd = psi(sampleOffdConfigs)
Eloc = H.get_O_loc(sampleLogPsi, sampleLogPsiOffd)
```

Besides the abstract `Operator` class that serves as a basis to implement arbitrary operators, the jVMC codebase provides two derived classes for typical applications:

- `BranchFreeOperator`: This class provides functionality to compose many-body operators as tensor products of branch-free operators acting on local Hilbert spaces. The term "branch-free" means that the local operators contain only a single non-zero entry in each row/column. An example are the Pauli operators, which are pre-defined as part of the module.

- `POVMOperator`: The `POVMOperator` class comprises utility to construct time-evolution operators in the POVM-formalism. Note that the computation of observables differs in this formalism, such that the returned operator may only be interpreted as the object generating the real time-evolution.

The documentation [39] explains the corresponding API in detail.

### 3.2.3   Sampler

The purpose of the `sampler` module is to handle all sampling-related tasks in a unified manner. The module provides two classes, the `ExactSampler` and the `MCSampler`. The `ExactSampler` evaluates the network on all the (exponentially many) basis configurations. Obviously, there is no associated computational speedup with this method and its main purpose is to make troubleshooting easier when testing new functionalities. In contrast, the `MCSampler` may be used to generate MC samples from the Born distribution of $\psi_\theta(\mathbf{s})$ either by direct sampling from an autoregressive NQS or by Metropolis-Hastings MCMC.

The common interface of both sampler classes is a `sample` member function, which generates a sample in the respective manner and returns a tuple of the generated basis configurations, the corresponding coefficients $\log \psi_\theta(\mathbf{s})$, and the probabilities $|\psi_\theta(\mathbf{s})|^2$ in the case of the `ExactSampler` or `None` in the case of the `MCSampler`. Hence, after instantiating a sampler object called `mySampler`, generating a set of samples amounts to the line

```
sampleConfigs, sampleLogPsi, sampleProb = mySampler.sample()
```

At instantiation both sampler types need to be passed the variational wave function and the sample shape. The `ExactSampler` furthermore requires the local Hilbert space dimension `lDim`.

For the initialization of a `MCSampler` a `jVMC.random.PRNGKey` is an additional necessary argument. An `updateProposer` function needs to be provided for MCMC sampling. The expected signature is `updateProposer(key, config, **kwargs)` and the function is supposed to return an updated basis configuration that is used as proposed move in the MCMC algorithm. If the optional argument `updateProposerArg` is given when instantiating a `MCSampler`, its value is passed to the `updateProposer` as `kwargs`. Further initialization arguments are the number of samples to generate `numSamples`, and for MCMC sampling the number of update proposals per sweep `sweepSteps` and the number of sweeps used for initial thermalization ("burn-in") `thermalizationSweeps`. The argument `numChains` defines the number of MCMC chains that are run in a vectorized manner in order to enhance the computational efficiency.

If the NQS which was given during the initialization of the `MCSampler` is an autoregressive model, i.e., if a `sample` member function exists in the respective Linen module, direct sampling is automatically used instead of MCMC sampling.

In both the `ExactSampler` and the `MCSampler` the generation of samples is automatically distributed across MPI processes and locally available devices. This means for MC sampling that when running $N_P$ MPI processes and calling the `sample` member function to obtain $N_{\mathrm{MC}}$ samples, each of the processes will produce $N_{\mathrm{MC}}/N_P$ samples. Due to the internal vectorization the total number of samples produced by the `MCSampler` can slightly exceed the number of samples asked for in order to match array dimensions.

### 3.2.4 MPI wrapper

It's amenability to parallelization across distributed compute clusters is a key feature of VMC algorithms, see Section 3.1. After generating samples locally on the different nodes, the results have to be collected in order to compute the quantities of interest. The `mpi_wrapper` module provides a system for the distributed sampling and the subsequent reduction tasks. For this purpose, it wraps the required MPI communications, for which the `mpi4py` package [37, 38] is used.

The function `jVMC.mpi_wrapper.distribute_sampling` distributes sampling tasks across the available processes and devices. For a desired total number of samples this function determines how many samples should be generated by each sampling process. The `ExactSampler` or `MCSampler` class invoke this function to automatically distribute the sampling tasks.

Assume that `mySampler` is an instance of a sampler class, `psi` is a variational quantum state, and `op` is an instance of a class derived from the `Operator` class, associated with a quantum operator $\hat{O}$. Then we can get samples and the corresponding $O^\theta_{\mathrm{loc}}(\mathbf{s})$ via

```
>>> s, logPsi, _ = mySampler.sample()
>>> sPrime, _ = op.get_s_primes(sampleConfigs)
>>> logPsiOffd = psi(sPrime)
>>> Oloc = op.get_O_loc(logPsi, logPsiOffd)
```

Now, on each MPI process `Oloc` is a two-dimensional array of size (number of devices) × (number of samples per device). On this basis, the `mpi_wrapper` module contains different functions to evaluate the distributed data. To get, for example, the Monte Carlo estimate of the expectation value of $\hat{O}$ as defined in Eq. (5), we can use the `get_global_mean` function

```
>>> Omean = jVMC.mpi_wrapper.get_global_mean(Oloc)
```

Further options are `get_global_variance`, `get_global_covariance`, and `get_global_sum` to compute the variance, co-variance matrix, and the sum of all elements, respectively.

### 3.2.5 Utilities

The `jVMC.util` module comprises higher-level code which is not at the heart of the jVMC codebase. The code in this module

- either contains functionality to combine the workings of the codebase on an intermediate (e.g., `jVMC.util.tdvp`, `jVMC.util.stepper`) or higher (e.g., ground state search or network initialization in `jVMC.util.util`) level,

- or solves tasks that are not immediately related to the core functionality (e.g. IO-tasks in `jVMC.util.output_manager` or symmetry transformations in `jVMC.util.symmetries`).

**TDVP**   The `jVMC.util.tdvp` module is centered around solving the equation

$$[[S_{k,k'}]]\dot{\theta}_{k'} = -[[\gamma F_k]] \;, \tag{40}$$

where $\gamma \in \{1, i\}$ is either the real or the imaginary unit. The module comprises a `TDVP` class with a `__call__` member function that returns a (regularized) solution $\dot{\theta}$ of Eq. (40), which corresponds to an SR step, Eq. (10), for $\gamma = 1$ or a real time step following the TDVP equation (19) for $\gamma = i$. The signature of the `__call__` method matches the standard right hand side required by ordinary differential equation solvers of the `scipy` library [62] or those provided as part of the jVMC codebase in the `jVMC.util.stepper` module (see below). Hence, an instance of the `TDVP` class can be passed as right hand side to these integrators. Invoking the `__call__` member function of a `TDVP` object, essentially amounts to carrying out the steps of Algorithm 1.

For solving Eq. (40) the `TDVP` class offers a number of options, of which we highlight a few, while referring to the detailed documentation [40] for a complete list:

- Choose $\gamma$ by setting the `rhsPrefactor` at initialization.

- Choose $[[\cdot]]$ to be $\mathrm{Re}(\cdot)$ or $\mathrm{Im}(\cdot)$ by passing `"real"` or `"imag"` for the argument `makeReal` at initialization.

- Set the regularization parameters $\nu$, $\epsilon_{\mathrm{pinv}}$, and $\epsilon_{\mathrm{SNR}}$ described in Section 2.5. The respective regularization scheme is not effective if the corresponding parameter is set to zero.

- The `hamiltonian` argument may be an `Operator` object or a function that takes a real number as argument and returns an `Operator` object. The latter can be used to solve the TDVP equation with time-dependent Hamiltonians.

- We observed that the `CUDA` implementation of the diagonalization can be unstable if the (quantum) Fisher matrix $S$ is ill-conditioned. To circumvent this issue, we introduced the `diagonalizeOnDevice` initializer argument, which allows to choose where to perform the diagonalization of the matrix.

- If the `__call__` function receives the optional `intStep` keyword argument, and if `intStep==0`, various quantities like energy mean and variance, $S$ and $F$, as well as residuals and the TDVP error of the solution are stored. These quantities can subsequently be retrieved via the corresponding `get_*` member functions of the `TDVP` class.

**Stepper**   The `stepper` module comprises two classes for the integration of ordinary differential equations: A simple `Euler` integrator and an `AdaptiveHeun` integrator. Both classes contain a `step` member function as common interface. Assuming that the ODE is given in the standard form $\dot{y}(t) = f(y(t), t, p)$, the `step` function takes $t_n$, $f$, $y(t_n)$, and optional additional parameters $p$ as argument and returns $y_{n+1} \equiv y(t_{n+1})$ following the respective integration scheme.

The `AdaptiveHeun` stepper is a second-order consistent adaptive integrator that adjusts the integration step size $\tau$ to achieve a given accuracy $\epsilon_{\text{step}}$. For this purpose, it computes solutions $y_{n+1}^{(\tau)}$ and $y_{n+1}^{(\tau/2)}$ using a Runge-Kutta scheme with two different step sizes $\tau$ and $\tau/2$. Based on the difference of the solutions $\delta = \left\| y_{n+1}^{(\tau)} - y_{n+1}^{(\tau/2)} \right\|$ the time step is adjusted according to $\tau \leftarrow \tau(\epsilon_{\text{step}}/\delta)^{1/3}$ in order to guarantee the desired accuracy.

One important point to observe is that the choice of the normalization measure, which is in principle arbitrary, can have a severe impact on the performance (i.e., the resulting step size). The norm can be chosen by passing a function that computes it as `normFunction` argument to the `step` function. A natural choice is to be sensitive only to deviations which correspond to actual changes in the wave-function. This is given, when using the (quantum) Fisher norm $\|\mathbf{v}\|_S = \frac{1}{N_p} \sqrt{\sum_{k,k'} v_k^* S_{kk'} v_{k'}}$, where $N_p$ is the number of components of $\mathbf{v}$ [24].

**Output Manager**   The output manager handles all IO-related tasks in a unified fashion. For output to a file it relies on the `hdf5` data-format which is handled using the `h5py` library [63]. During evolution, three different types of data are typically stored:

- physical observables, corresponding to the `write_observables` function expecting a timestamp and a dictionary of to-be-written observables,

- metadata, corresponding to the `write_metadata` function expecting a timestamp and a dictionary of metadata to be written,

- network checkpoints, corresponding to the `write_network_checkpoints` function expecting a timestamp and the set of network parameters at that time step. This allows to restart the simulation at arbitrary times or to sample observables which have not been stored during the initial run, using the `get_network_checkpoint` function to set desired network parameters for all MPI-processes.

Moreover, the `OutputManager` class provides a `print` function for printing to the standard output, which blocks all MPI processes except for the root process to avoid repeated output.

**Symmetries**   Incorporating symmetries in the NQS turned out essential for high accuracy in various applications. For the symmetrization a typical task is to produce the symmetry-transformed input configurations $s_{\omega(l)}$ for all transformations $\omega \in \Omega$; see, e.g., Section 5.1 or Refs. [10, 17].

The `jVMC.util.symmetries` module contains functions that return the complete sets of symmetry transformations $\Omega$ for simple lattices in the form of permutation matrices. These can be used in definitions of NQS architectures to produce all symmetry-related configurations by simple dot-products, see the code of symmetrized architectures in the `jVMC.nets` module for examples.

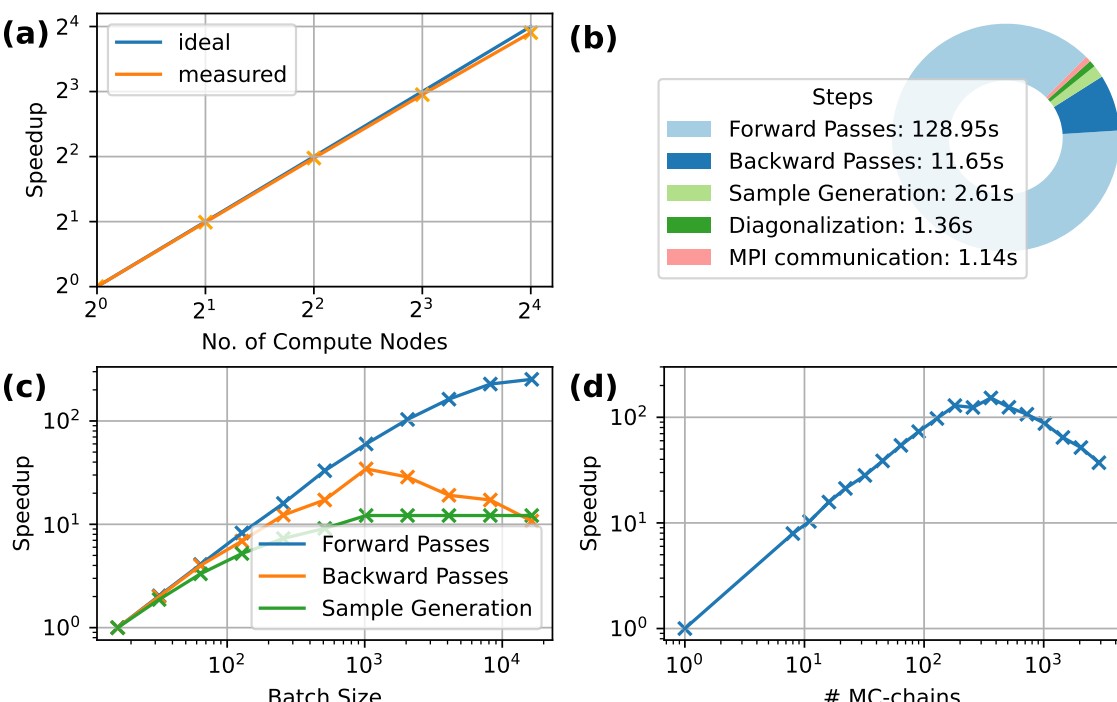

Figure 3: (a) Scaling of the computational runtime with the number of compute nodes. Here, each node is equipped with `NVIDIA A100` GPUs. (b) Execution times of the different tasks in a single step in case of a single node. The benchmark script is available at `vmc_jax/examples/ex4_benchmarking.py`. (c) Speedup gained by increased arithmetic intensity via batching. (d) Effect of multiple parallel MCMC-chains on the runtime.

## 4 Performance

One of the key features of the jVMC codebase is the efficient exploitation of large scale supercomputing resources for NQS simulations. Ideally, many accelerators (GPUs or TPUs) are available to share the independent workload for each batch of samples as described in Section 3.1.

Fig. 3a demonstrates close to ideal speedup achieved by the MPI parallelization of the code. As a benchmark we consider an SR step for a 50-site transverse-field Ising chain with periodic boundary conditions. The NQS architecture is a symmetrized RNN with 1510 variational parameters and the expectation values are estimated using $N_{\mathrm{MC}} = 3{\times}10^5$ Monte Carlo samples. The displayed timings were taken on the `JUWELS Booster` module at the Jülich Supercomputing Centre [64], where each node is equipped with four `NVIDIA A100` GPUs. Using the single-node performance as baseline, we observe almost ideal speedup for up to 16 nodes or 64 GPUs. The breakdown of compute times on a single node in Fig. 3b shows that network evaluations account for the majority of the computational cost, which explains the efficiency of the parallelization. Notice the negligible contribution of the sample generation, which is due to direct sampling from the autoregressive NQS.

Additionally, we benchmark the performance gain obtained when increasing the batch size and, thereby, the arithmetic intensity of the network evaluation tasks. As discussed in Section 3.1.2, the batch size should ideally be chosen to saturate the GPU's memory capacity. To illustrate the significance of suited batching, we again time one SR update step, for this purpose considering a 30-site spin chain and an RNN without symmetries. For $N_{\mathrm{MC}} = 10^4$, the speedup gained by increasing the batch size is shown in Fig. 3(c). The

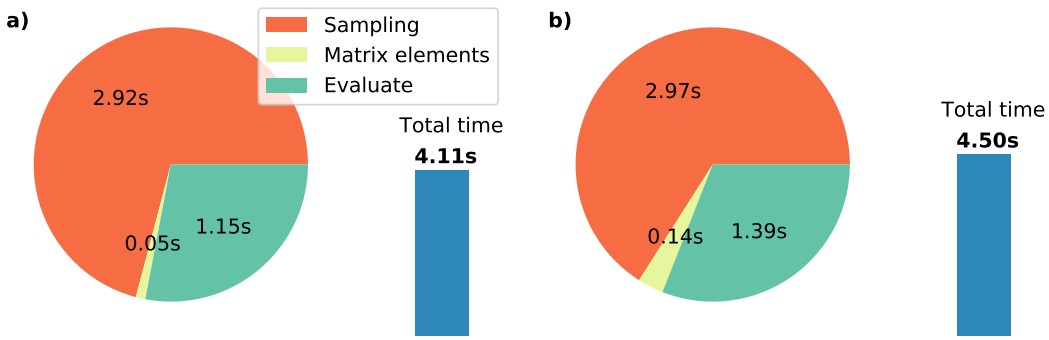

Figure 4: Performance comparison between (a) the jVMC codebase and b) the NetKet library [65]. The timings were performed on a `Intel Xeon Gold 6240 2.6GHz` CPU with an attached `NVIDIA Tesla V100` GPU.

acceleration of the forward pass by more than a factor of 100 compared to the baseline with batch size 16 underlines the importance of batching for optimal performance.

The performance gain by batching is also the reason why running multiple parallel Monte Carlo chains can enhance the performance. However, there exists a sweet spot in the number of chains as shown in Fig. 3(d). For very few MC-chains the GPU is not used to its full capacity as the network is evaluated on too small batches. In the opposite limit, we attribute the deteriorating performance to a saturation of the occupation of the Streaming Multiprocessors such that computations start to be performed sequentially. This leads to effective idle times of the individual chains and therefore to degraded overall performance. Fig. 3(d) shows optimum performance with of the order of 100 parallel Monte Carlo chains, which we found to be a good number in various simulations; here, we used a one-dimensional complex Convolutional Neural Network with an input size of $N = 80$ and four subsequent layers of sizes $10, 8, 6, 4$ with a filter diameter of 20 (6536 variational parameters). The sample size was $N_{\mathrm{MC}} = 10^4$ and the MCMC was performed with sweep size $N$ and 20 initial thermalization sweeps. These simulation parameters are again used for the following test case.

Finally, we include a direct comparison to the NetKet library [65]; notice that NetKet version 3.2, which we are using for this comparison, relies mostly on JAX for the computationally intense parts of the code – this recent development is not described in Ref. [65]. As a benchmark problem we choose the evaluation of an operator expectation value, i.e., the evaluation of Eq. (5), whose cost is comparable to the computation of the energy gradient $\partial_{\theta_k} E(\theta)$. As operator we use the transverse-field Ising Hamiltonian with system size $N = 80$. The NQS architecture is the same as in Fig. 3(d) and the sample of size $N_{\mathrm{MC}} = 10^4$ is generated using the approximately optimal number of 350 parallel MCMC chains. We separate the timing into three contributions, (i) the sampling, (ii) the computation of the matrix elements, and (iii) the final evaluation of Eq. (5), which includes the evaluation of $\psi(\vec{s})$ on the involved configurations $\mathbf{s}$ and $\mathbf{s}'$. The resulting runtimes are shown in Fig. 4(a) for jVMC and in Fig. 4(b) for NetKet. For the given task jVMC is in total about 10% faster than NetKet. The difference between both codes lies in the computation of matrix elements and the final evaluation. To our understanding the difference in runtimes originates in the fact that in jVMC all computational operations are performed on the GPU, while NetKet performs the computation of matrix elements on the CPU. Hence, the parallelism of this task is not fully exploited in NetKet and additional overheads are created due to data transfer between CPU and GPU.

## 5 Examples

### 5.1 Ground state search

For the exemplary ground state search we consider the one-dimensional transverse-field Ising model

$$\hat{H} = -\sum_l \hat{Z}_l \hat{Z}_{l+1} - g \sum_l \hat{X}_l \tag{41}$$

with Pauli operators $\hat{X}_l$ and $\hat{Z}_l$ acting on lattice site $l$ and periodic boundary conditions. Fermionization of the Hamiltonian results in a single particle problem, which can be solved efficiently for large finite systems [66]. In the following, this is our reference for the exact ground state energy.

The un-normalized wave function ansatz is a fully connected single-layer CNN with real parameters, meaning that the NQS automatically incorporates the translational symmetry of the Hamiltonian:

$$\log \psi_\theta(\vec{s}) = \sum_{a=1}^{\alpha} \sum_{\omega \in \mathcal{T}} f_{\text{ELU}}\big(W_{al} s_{\omega(l)} + b_a\big) \tag{42}$$

Here, $\mathcal{T}$ is the set of translations and $f_{\text{ELU}}(x)$ denotes the ELU activation function

$$f_{\text{ELU}}(x) = \begin{cases} x & \text{if } x > 0 \\ e^x - 1 & \text{if } x \leq 0 \end{cases} . \tag{43}$$

The ground state search is performed by Stochastic Reconfiguration, see Section 2.2, using the learning rate $\tau = 10^{-2}$, $N_{\text{MC}} = 4 \times 10^4$ MC samples, an initial regularization with diagonal shift $\nu_0 = 10$ that decays with increasing step number $n$ as $\nu = 0.95^n \nu_0$, and a pseudo-inverse cutoff parameter $\epsilon_{\text{pinv}} = 10^{-8}$ (see Section 2.5).

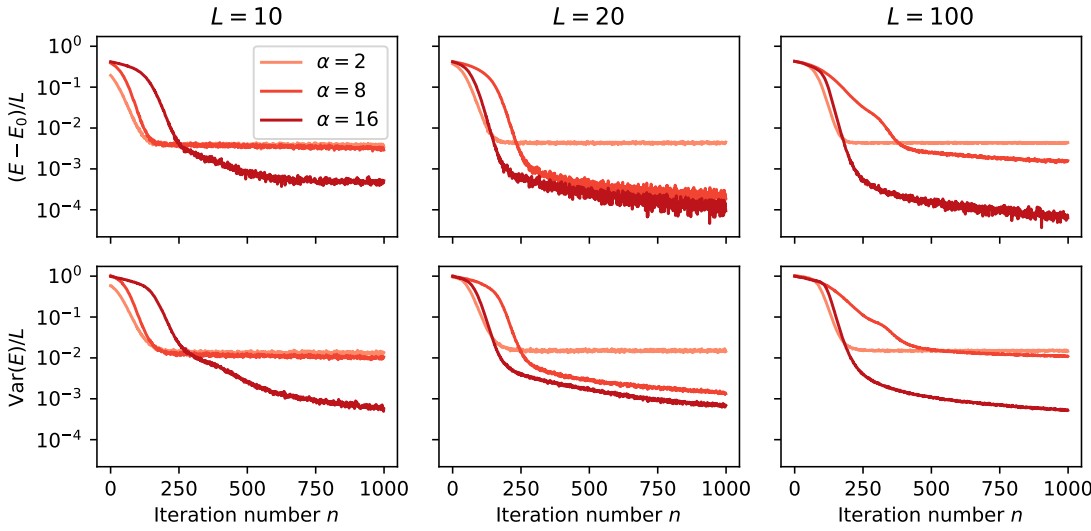

Figure 5: Example ground state search for the transverse-field Ising model using NQS and Stochastic Reconfiguration for different network sizes $\alpha$ and system sizes $L$. The top row shows the deviation from the exact reference energy $E_0$. The bottom row shows the corresponding energy variance, which vanishes in an exact eigenstate.

Fig. 5 shows the evolution of energy and energy variance during the optimization for different system sizes $L = 10, 20, 100$ and $g = 0.7$. The parameter $\alpha$ controls the size of the NQS (see Eq. (42)) and thereby the quality of the result.

This basic ground state search is implemented in the example notebook `examples/ex0_ground_state_search.ipynb`, which is contained as an example in the codebase repository and reproducing them is feasible on a single GPU.

## 5.2 Real time dynamics

### 5.2.1 Unitary dynamics of pure states

As an example of unitary dynamics we simulate a quench in the transverse-field Ising model on a square lattice

$$\hat{H} = -J \sum_{\langle i,j \rangle} \hat{Z}_i \hat{Z}_j - g \sum_j \hat{X}_j \tag{44}$$

where $\hat{X}_j$ and $\hat{Z}_j$ are Pauli operators and $\langle i, j \rangle$ denotes pairs of nearest neighbors on the lattice. The initial state $|\psi_0\rangle$ is the paramagnetic ground state at $J = 0$, which we find by an initial ground state search. Subsequently, the system evolves with the Hamiltonian parameters $J = 1$ $g = 1.5g_c$, where $g_c = 3.04438J$ [67] is the critical transverse field strength at which the model exhibits a phase transition. We consider open boundary conditions, meaning that the physical setting is a variant of the situation addressed in [24].

The network architecture of the NQS is a single-layer fully connected complex CNN with suited zero-padding of the computational basis configuration to account for the open boundary conditions. The activation function is the Taylor series of $\log \cosh(x)$ truncated at sixth order. For a system of $8 \times 8$ lattice sites we show in Fig. 6a) the time evolution of the transverse magnetization at a central site, $\langle \hat{X}_{L/2,L/2} \rangle$, obtained with different network sizes to demonstrate convergence, namely 8, 16, 24, and 32 channels in the CNN. Fig. 6b) displays the spatio-temporal spreading of correlations based on the correlation function $\langle \hat{Z}_{L/2,L/2} \hat{Z}_{i,j} \rangle$.

This result was obtained using $N_{\text{MC}} = 10^6$ Monte Carlo samples. The TDVP equation was solved using cutoff parameters $\epsilon_{\text{pinv}} = 10^{-8}$ and $\epsilon_{\text{SNR}} = 2$. The tolerance for the adaptive time step of the integrator was $\epsilon_{\text{step}} = 10^{-6}$.

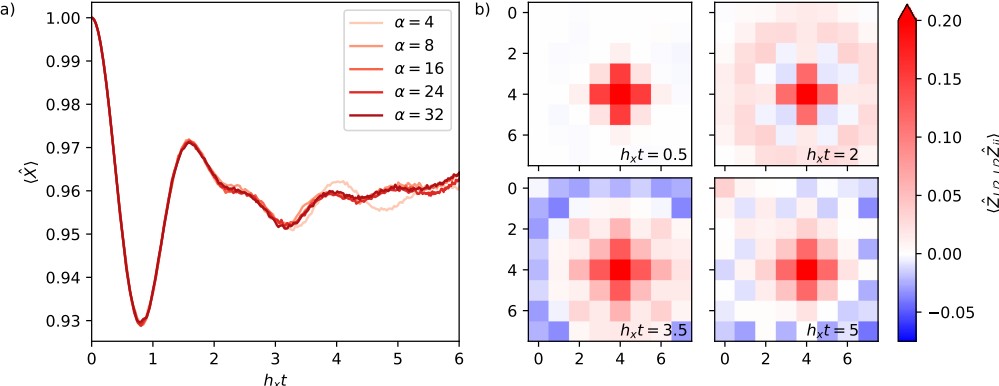

Figure 6: Quench dynamics in a two-dimensional quantum Ising model of size $8 \times 8$. (a) Time evolution of the transverse magnetization at the center of the lattice. (b) Spatio-temporal spreading of correlations.

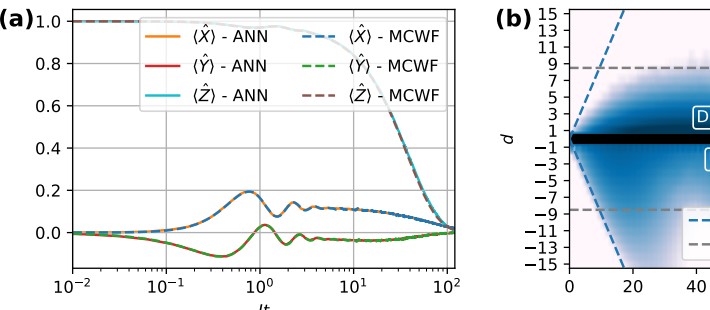

Figure 7: (a) Mean magnetizations in a spin chain of length $N = 32$ with the quench parameters $h_x/J_z = 0.25$, $h_z/J_z = 0.05$ and the dissipation channel $\hat{L} = \hat{Z}$ with relative strength $\gamma/J_z = 0.25$ compared to MCWF-data for $N = 16$ spins starting in the product state $\langle \hat{Z} \rangle = 1$. (b) Spreading of correlations in the spin chain. Top: Dissipative system with $\gamma = 0.25$, Bottom: MPS simulation of the unitary system where $\gamma = 0.0$. After an initial linear light-cone spreading, the nature of the dissipative propagation grows more diffusive, before all correlations eventually vanish. Notice that the slight deviations in panel (a) coincide with the time at which the dissipative correlations cross the MCWF system size boundary. Data taken from Ref. [29].

### 5.2.2 Dissipative dynamics

Different paths to variational approximations of $\hat{\rho}$ for open system dynamics using neural networks have been explored [27–29], which were briefly introduced in Section 2.4. Here, we showcase results that were obtained in Ref. [29] employing the probability based POVM approach in conjunction with the jVMC codebase. We study a quench of a $z$-polarized state in a one-dimensional quantum Ising model with transverse and longitudinal field; the openness consists of dephasing with relative strength $\gamma/J = 0.25$, which eventually drives the system to the featureless steady state $\hat{\rho}_{SS} \propto \mathbb{1}$. The system's Hamiltonian is accordingly given by

$$\hat{H} = \sum_{\langle ij \rangle} J_z \hat{Z}_i \hat{Z}_j + \sum_i \left( h_z \hat{Z}_i + h_x \hat{X}_i \right) . \tag{45}$$

and is of interest as it exhibits confinement effects which may be regulated by tuning $h_z$ [68]. Non-zero values of $h_z$ break the $Z_2$ symmetry of the TFIM-Hamiltonian and function as an energy penalty for domains that point in opposite direction from $h_z$, thereby suppressing the spreading of correlations. The obtained results are shown in Fig. 7.

Here, an RNN with 5 layers was used where in each layer, latent information is carried in the form of a hidden state of length 12. The corresponding transformations are described by 1456 variational parameters, which are updated using $N_{\mathrm{MC}} = 1.6 \times 10^5$ samples in each time step with a prescribed integration tolerance $\epsilon_{\mathrm{step}} = 10^{-3}$ and $\epsilon_{\mathrm{pinv}} = 10^{-8}$. The SNR regularization scheme was not used and translational invariance was enforced as described in Sec. 3.2.5.

## 6 Conclusion

The jVMC codebase provides a basic and flexible framework for the composition of NQS algorithms. The typical building blocks – on-the-fly computation of operator matrix elements, sampling from the Born distribution, and evaluating the wave function ansatz and its gradients – are implemented in an efficient manner. While designed to fully exploit the

resources of distributed compute clusters, the underlying just-in-time compilation allows the user to run the identical Python code just as well on a desktop computer without accelerators.

Compared to the existing NetKet library [65], the jVMC codebase was devised following a different philosophy. While NetKet provides implementations of many different NQS algorithms, network architectures, and physical models on top of the core building blocks, the jVMC codebase focuses on exposing a minimal system of efficiently implemented modules that facilitates the custom composition of such algorithms. This bottom-up ansatz is intended to free future method development from the necessity of first having to create parallelized and performant implementations of the core tasks; at the same time we aimed to impose only minimal bias with respect to the details of resulting algorithms.

The presented codebase has already been used in research applications to compute the energy gap of a two-dimensional quantum Ising model and its unitary dynamics across a quantum phase transition [25], and also for the simulation of open quantum spin systems in one and two dimensions [29]. We expect that it will serve as a useful basis for further method development and applications in these and other directions, where VMC with NQS can potentially push our classical simulation capabilities.

# Acknowledgements

**Funding information** MR was supported by the Deutsche Forschungsgemeinschaft (DFG, German Research Foundation) under Germany's Excellence Strategy EXC2181/1-390900948 (the Heidelberg STRUCTURES Excellence Cluster) and within the Collaborative Research Center SFB1225 (ISOQUANT); moreover, MR was partially supported by the Baden-Württemberg Stiftung gGmbH. The authors gratefully acknowledge the Gauss Centre for Supercomputing e.V. (www.gauss-centre.eu) for funding this project by providing computing time through the John von Neumann Institute for Computing (NIC) on the GCS Supercomputer JUWELS at Jülich Supercomputing Centre (JSC) [64]. Furthermore, the authors acknowledge support by the state of Baden-Württemberg through bwHPC and the German Research Foundation (DFG) through grant no INST 40/575-1 FUGG (JUSTUS 2 cluster).

# A  The underlying JAX library in a nutshell

The jVMC codebase is built on the JAX library, which provides automatic differentiation, vectorization, and just-in-time compilation to accelerators in order to enable "high-performance numerical computing and machine learning research" [36]. JAX can automatically differentiate and vectorize functions that are written in Python and it can compile the Python code for efficient execution on the CPU or on GPUs/TPUs if available. As a basis the JAX library contains an own implementation of (most of) the NumPy [69] functionality in the `jax.numpy` submodule. In the following we sketch the main components of a system of composable function transformations, namely just-in-time compilation with `jax.jit`, automatic differentiation with `jax.grad`, vectorization with `jax.vmap`, and parallelization across multiple accelerators with `jax.pmap`.

Importantly, the ability to perform the various function transformations imposes a number of constraints on how those functions are written. Here, we highlight the following:

- **Pure functions:** All input data have to be passed to the function as arguments and all results given as return values. Global variables and side effects lead to unexpected

behavior of JAX-transformed functions. Some workarounds are needed to reconcile JAX with object-oriented code.

- **Control flow** needs some care. Replace Python `for`-loops or `if`-branching with primitives from the `jax.lax` sub-module, see the JAX documentation for details.

- **JAX arrays are immutable** in order to enable function tracing. However, when attempting to change the content of a JAX array, the thrown error gives hints to the correct pure analog of this operation.

```
>>> jax_array = jax.numpy.ones((3, 3), dtype=jax.numpy.float32)
>>> jax_array[1, :] = 2.0
TypeError: '<class 'jaxlib.xla_extension.DeviceArray'>' object does not
    support item assignment. JAX arrays are immutable; perhaps you want
    jax.ops.index_update or jax.ops.index_add instead?
```

  One additional option besides the suggestions to resolve the issue is

```
>>> jax_array.at[1, :].set(2.0)
```

For a comprehensive list, we refer the reader to the section about "the sharp bits" of the JAX documentation [70]. The example codes below are variations of examples shown in the JAX documentation.

## A.1 Just-in-time compilation

`jax.jit` returns a compiled version of the function that is given as the argument. Importantly, the function will be recompiled whenever the input data type or shape is changed.

```python
def selu(x, alpha=1.67, lmbda=1.05):
    return lmbda * jax.numpy.where(x>0, x, alpha * jax.numpy.exp(x)-alpha)

selu_jitd = jax.jit(selu)

x = jax.numpy.arange(-2,2,1)
print(selu_jitd(x)) # [-1.5161895 -1.1084234 0. 1.05]
```

When running this piece of code, the function will first be compiled when the last line is executed and then the compiled function will be called. In this example, calling the jitted function instead of the original Python function will not exhibit any significant differences (besides a compilation overhead when calling the jitted function for the first time). Compiling functions that execute considerable computational workload, instead, can yield significant gains in performance.

`jax.jit` can be used to showcase an example of unexpected behavior when global variables are involved in JAX-transformed functions:

```python
>>> global g
>>> g = 3
>>> def f(x):
...     return x + g
>>> f_jit = jax.jit(f)
>>> f_jit(0)
DeviceArray(3, dtype=int32)
>>> g = 4
>>> f_jit(0)
DeviceArray(3, dtype=int32)
```

```
>>> f_jit(0.)
DeviceArray(4., dtype=float32)
```

As mentioned above, a jitted function is recompiled if invoked with new data types, which is the reason for the output of the last line.

## A.2 Automatic differentiation

The restriction to pure functions is the basis for an easily accessible implementation of automatic differentiation in JAX. The derivative of arbitrary pure functions can be computed with `jax.grad`. Moreover, the function returned by `jax.grad` can be used in further transformations, for example to compute the second derivative in the code below. If derivatives for more than the first argument are required, the keyword-argument `argnums` needs to be specified.

```python
# Define a function
def f(x):
    return x**2

df = jax.grad(f) # Get the gradient of the function
ddf = jax.grad(df) # Get the second gradient of the function

x_list = jax.numpy.arange(0,1,.2)
print(jax.numpy.array([df(x) for x in x_list])) # [0. 0.4 0.8 1.2 1.6]
print(jax.numpy.array([ddf(x) for x in x_list])) # [2. 2. 2. 2. 2.]
```

## A.3 Vectorization

To accelerate typically slow Python `for`-loops and in order to enable the full exploitation of accelerators, JAX comprises can transform a *single-instruction single-data* (SISD) operation (like the function `f` defined above) into a function that works on multiple data (*single-instruction multiple-data*, SIMD). For a given function that operates on some input data, `jax.vmap` returns a vectorized version of that function, that applies the original function *element-wise* to vectors of input data.

As mentioned above, all JAX function transformation are composable, meaning that one can arbitrarily combine `jax.jit`, `jax.grad`, and `jax.vmap` as demonstrated in the following example:

```python
# Define a function
def f(x):
    return x**2

df = jax.vmap(jax.grad(f)) # Get the vectorized gradient of the function

x_list = jax.numpy.arange(0,1,.2)
print(df(x_list)) # [0. 0.4 0.8 1.2 1.6]
```

## A.4 Parallelization across multiple accelerators

When multiple accelerators are available, `jax.pmap` can map a function to these devices, such that the function is executed in parallel with the elements along one axis of the input data as arguments. Thereby, `jax.pmap` is superficially very similar to `jax.vmap`. But, in fact, the effect of both transformations is quite different: While `jax.vmap` vectorizes

a function, `jax.pmap` creates copies of the function which are then carried out on each device individually. This means, for example, that the data that is passed to a pmapped function has to be physically scattered across different devices.

There are some pitfalls that one needs to be aware of; one example is that the leading axis dimension of inputs must not be greater than the number of devices. A practical way of circumventing such issues and to obtain good parallelization results is to reshape the leading axis of datasets into the number of devices, which can be obtained using `jax.local_device_count()`. In the example below we assume that two GPUs are available.

```python
# Define a function
def f(x):
    return x**2

df = jax.pmap(jax.grad(f)) # Get the vectorized gradient of the function

x_list = jax.numpy.arange(0,1.2,.2).reshape((jax.local_device_count(), -1))
print(df(x_list)) # [[0. 0.4 0.8], [1.2 1.6 2.0]]
```

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
