# Peer review of "jVMC: Versatile and performant variational Monte Carlo leveraging automated differentiation and GPU acceleration"

_SciPost Physics Codebases, doi:SciPost Phys. Codebases 2 (2022) , SciPost Phys. Codebases 2-r0.1 (2022)_

## Round 2 · Referee Report · Everard van Nieuwenburg (Referee 1) · 2021-12-21

Report

Dear Authors,

The updates you have made to the paper take into account the comments and suggestions I made. I am especially happy to see:

  • An extra figure (Fig 2) showing how the modules are related.
  • A timing comparison between this code and NetKet, which --even though this code provides a different (lower level) user experience-- provides related capabilities for some of the benchmarks you include.

Though I do not believe it necessary to suggest this as a change to the paper, I do wonder about the timing comparison: At first sight it may seem like the faster matrix element computation (which you suggest comes from CPU vs GPU overhead) is irrelevant for large amounts of samples. The latter will quickly dominate the runtime. You mention that the sampling itself was implemented using a near optimal choice of 350 parallel chains for MCMC, but that is irrespective of the nr of samples, right?

Do not get me wrong: even if both packages are equally fast, they both have their own (dis-)advantages that in no way diminishes the usefulness of either. I also did not take into account whether more samples (than 10^4) are typically necessary, in which case the 10% increase in runtime is absolutely valid.

Regardless of the above, I think the paper has improved since the first version and is now even more accessible for newcomers.

---

## Round 2 · Referee Report · Anonymous (Referee 2) · 2021-12-28

Strengths

1) The revised manuscript provides more details on the SR optimization method and on the choice of regularization technique. 2) The revised manuscript includes a performance comparison against an analogous computational library, namely, NetKet.

Weaknesses

1) As the authors point out, more research is needed to fully establish which regularization method is to be preferred.

Report

The authors have exhaustively addressed all clarifications I requested in my previous report. The revised manuscript meets the criteria for publication in SciPost Physics Codebases.

---

## Round 2 · Author Response

Dear Prof. Corboz,

We are resubmitting a revised version of our manuscript. For the revision we addressed all comments and suggestions by the Referees. We provide our response to their reports separately as "Reply to the report" on the website. Below, we summarize the changes in the new version of the manuscript.

Thank you for pointing out the issue with `partial` in the new version of JAX. We fixed it in a new version of the code.

Sincerely,
Markus Schmitt and Moritz Reh

---

## Round 2 · List of Changes

• In the introduction we replaced the term “intermediate spatial dimensions” by “two or three spatial dimensions”.
  • We rephrased the sentence between Eq. (10) and Eq. (11).
  • We moved the pseudocode (Algorithm 1) to the end of Section 2.3 and expounded more clearly the analogy between unitary time evolution and SR.
  • We renamed section 2.4 to “Markovian dissipative dynamics” and rewrote large parts of it for clarification.
  • We rephrased the original formulation "...more forgiving in this regard due to its projective nature" in Section 2.5.1 to better clarify what we mean.
  • We added a paragraph to Section 2.6.2 discussing the computational cost of autoregressive sampling vs. MCMC.
  • We added Figure 2 with an overview of the core modules.
  • We added a performance comparison with NetKet in Section 4.
  • We added references [31,32].
  • We expanded the introductory paragraph of Section 2.5 with a brief outline of the regularization approaches.
  • We added the visible biases to the definition in Eq. (37) and the corresponding example code.
  • We changed the explanation after Eq. (40) to clarify what the exact reference is.
  • We included a list with references to exemplary ANN architectures below Eq. (32).
  • We updated Fig. 6 due to a mistake in the original figure.
  • We updated a number of references that have been published in the meantime.
  • In Section 4 we updated our explanation for the deteriorating performance in the limit of many parallel MC chains shown in Fig. 3d), because we realized that our previous understanding in terms of the thermalization overhead was not correct.

---

## Editorial Decision

published